# Resistant and Susceptible *Pinus thunbergii* ParL. Show Highly Divergent Patterns of Differentially Expressed Genes during the Process of Infection by *Bursaphelenchus xylophilus*

**DOI:** 10.3390/ijms241814376

**Published:** 2023-09-21

**Authors:** Tingyu Sun, Mati Ur Rahman, Xiaoqin Wu, Jianren Ye

**Affiliations:** 1College of Forestry, Nanjing Forestry University, Nanjing 210037, China; suntingyunjfu@163.com (T.S.); mati@njfu.edu.cn (M.U.R.); xqwu@njfu.edu.cn (X.W.); 2Collaborative Innovation Center of Sustainable Forestry in Southern China, Nanjing 210037, China

**Keywords:** *Pinus thunbergii*, pine wilt disease, resistant to nematode, somatic plantlets, transcriptome

## Abstract

Pine wilt disease (PWD) is a devastating disease that threatens pine forests worldwide, and breeding resistant pines is an important management strategy used to reduce its impact. A batch of resistant seeds of *P. thunbergii* was introduced from Japan. Based on the resistant materials, we obtained somatic plants through somatic embryogenesis. In this study, we performed transcriptome analysis to further understand the defense response of resistant somatic plants of *P. thunbergii* to PWD. The results showed that, after pine wood nematode (PWN) infection, resistant *P. thunbergii* stimulated more differential expression genes (DEGs) and involved more regulatory pathways than did susceptible *P. thunbergii*. For the first time, the alpha-linolenic acid metabolism and linoleic acid metabolism were intensively observed in pines resisting PWN infection. The related genes *disease resistance protein RPS2* (*SUMM2*) and pathogenesis-related genes (*PR1*), as well as reactive oxygen species (ROS)-related genes were significantly up-expressed in order to contribute to protection against PWN inoculation in *P. thunbergii*. In addition, the diterpenoid biosynthesis pathway was significantly enriched only in resistant *P. thunbergii*. These findings provided valuable genetic information for future breeding of resistant conifers, and could contribute to the development of new diagnostic tools for early screening of resistant pine seedlings based on specific PWN-tolerance-related markers.

## 1. Introduction

*Pinus thunbergii* Parl. was the most prestigious of the coniferous bonsai trees, which was a type of positive evergreen tree native to Japan and the southern coast of Korea [1,2] It was widely used in coastal urban greening and coastal windbreaks due to its resistance to barrenness and strong adaptability, especially to sea fog and sea winds [3,4,5,6]. It has been cultivated in Qingdao City, China, since 1900 and has been widely planted in coastal cities in China for almost 120 years [7,8]. However, the population of *P. thunbergii* has declined dramatically since the outbreak of pine wilt disease (PWD) [9]. PWD has severely limited the development of *P. thunbergii* forests.

PWD was a devastating forest disease of the *Pinus* species, caused by the pine wood nematode (PWN) *Bursaphelenchus xylophilus* [10]. It was first reported in 1905 in Nagasaki, Japan, as a serious forest disease affecting the *Pinus* species (*P. thunbergii* and *P. densiflora*) [11,12]. Now, PWD has been reported in the USA, Canada, Mexico, China, Japan, Korea, Portugal and Spain [13,14]. Since PWD became endemic in various countries, its spread has been beyond expectations and caused huge economic losses, especially in China and Japan [15,16,17]. Among the 353 species of Pinaceae, up to 106 *Pinus* species could be infected by PWN, and the host range of PWN is expanding due to its adaptability [18]. In addition to the wide host range, the hazard range of PWD has been gradually expanding due to global warming [19,20]. Nowadays, PWD has become the greatest threat to the development of pine forests worldwide [21,22]. The comprehensive understanding and control of PWD is a work-in-progress. Many methods have been investigated to control PWD, and although none has been proven to completely control the disease, they have been effective in slowing down the spread of PWD [23]. Within this holistic framework, the use of tolerant/resistant *P. thunbergii* cultivars was probably the most efficient and environmentally friendly approach to controlling PWD. Due to the extreme difficulty of preventing the disease, there has been a growing enthusiasm for processes of breeding pine trees for resistance to PWD. Several studies have focused on finding sources of genetic resistance to PWN [24] Japan was the first country to select resistant *Pinus* species from PWD-endemic areas [25,26] Our group introduced thirteen resistant families (Nos. 28–40) of *P. thunbergii* seeds in 2004, and the seedlings from the introduced resistant families showed PWD resistance after artificial inoculation [27]. Based on the resistant materials, many somatic plants were obtained by our group [28].

Plant resistance mechanisms are important for maintaining the survival and health of organisms against adverse environments and pathogen invasions [15,29,30]. Currently, there have been more than 2000 reports of pine resistance to PWN, but the mechanism of pine resistance to PWN is still inconclusive [16]. Identifying the mechanisms of pine resistance to PWN could help us to better understand the pathogenesis and develop precise prevention or control measures [31]. This would facilitate better control of PWD epidemics in the future. Pines have enormous genomes with large amounts of repetitive DNA, making whole-genome sequencing projects difficult [32,33]. Transcriptome analysis based on next-generation sequencing data (RNA-Seq) provides information on all transcriptional activity in a cell or organism [34]. For pines, RNA-Seq was an efficient method to generate functional genomic data due to the absence of a valid reference genome sequence [35]. To comprehensively understand the pathogenic mechanism and reduce the damage caused by PWD, the genetic analysis studies had been conducted after inoculating the pines with PWN, including such plants as *P. densiflora* [36], *P. pinaster* [37], *P. massoniana* [38] and *P. radiata* [39]. However, to our knowledge, although transcriptomic studies of PWD resistance in *P. thunbergii* had been reported, the information on PWN resistance genes in somatic plants was rare.

In this paper, we investigated for the first time the genetic information of somatic plants in response to PWN infection. The 3-year-old somatic plants of *P. thunbergii* were artificially inoculated with a highly virulent strain of PWN to test their resistance to PWD, and differentially expressed genes and metabolic pathways associated with PWN resistance were revealed by comparing the responses of susceptible and resistant *P. thunbergii* at the transcriptome level; candidate resistant genes that explained the different levels of PWN susceptibility were identified. Our characterization of the response to PWN infection would contribute to the future molecular breeding of PWN-resistant *P. thunbergii* and further understanding of the mechanisms of resistance to PWN inoculation in *Pinus* species.

## 2. Results

### 2.1. DEGs Vary in Response to PWN Inoculation of Susceptible and Resistant P. thunbergii

The resistant and susceptible *P. thunbergii* showed very different regulatory processes in response to PWN infection. In PWN-resistant *P. thunbergii*, the more DEGs (including up- and down-regulated genes) were triggered in response to PWN infection, e.g., 611 up- and 253 down-regulated DEGs responded to PWN infection in PWN-resistant *P. thunbergii* from 1 to 3 dpi, whereas only 21 up- and 46 down-regulated DEGs were induced in susceptible *P. thunbergii*. Furthermore, the DEGs’ evolutionary trends were different, with up-regulated DEGs predominating in PWN-resistant *P. thunbergii* during the primary stage of PWN infection (from 1 to 3 dpi), and up- and down-regulated genes fading in and out thereafter. The opposite was true for susceptible *P. thunbergii* (Figure 1A,B). During the different response processes to PWN infection, the common genes involved in both resistant and susceptible *P. thunbergii* were relatively few. For example, 10 common DEGs were found between 1d vs. 3d and 3d vs. 7d in susceptible *P. thunbergii*, and 73 common DEGs existed in PWN-resistant *P. thunbergii* (Figure 1C,D). The common DEGs for both susceptible (10/332) and PWN-resistant *P. thunbergii* (73/1783) were less than 5% of the total number. In addition, some specific DEGs were present in the PWN infection process. For example, 57 and 265 specific DEGs were found in 1 d vs. 3 d and 3 d vs. 7 d stage, respectively. The number of DEGs involved in the defense response of *P. thunbergii* varied considerably at different stages of PWN infection development. This suggested that the defense responses of resistant and susceptible *P. thunbergii* were quite different as the disease progressed.

### 2.2. Functional Annotation and Enrichment Analysis of DEGs

To gain a functional perspective of the DEGs associated with resistance to PWN, we performed a GO annotation analysis (*p* < 0.05) of these DEGs in resistant and susceptible phenotypes during the disease’s progression. Four DEG sets of two PWN infection stages during disease progression were subjected to GO annotation analysis to identify over-represented GO terms. We found that the vast majority of DEGs were different in both phenotypes and that fewer GO terms were activated in susceptible *P. thunbergii*, including biological process, cellular component and molecular function category, than in the PWN-resistant *P. thunbergii*, especially in the first infection stage (1 d vs. 3 d). Many GO terms were activated only in resistant *P. thunbergii,* such as: developmental process, multi-organism process, organelle part and so on (Figure 2A). Furthermore, we performed a KEGG enrichment analysis (*p* < 0.05) of these DEGs. These results also showed that the enriched pathways associated with the DEGs were more diverse in resistant *P. thunbergii*. Only two pathways (mitogen-activated protein kinase (MAPK) signaling pathway—plant, biosynthesis of various secondary metabolites—part 2) associated with the DEGs were enriched in the early infection stages (1 d vs. 3 d) of susceptible *P. thunbergii*, compared to eight in PWN-resistant *P. thunbergii*. And the alpha-linolenic acid metabolism and linoleic acid metabolism pathways were enriched in susceptible *P. thunbergii* at later infection stages (3 d vs. 7 d), compared to the seven pathways in PWN-resistant *P. thunbergii* (Table 1). The alpha-linolenic acid metabolism and linoleic acid metabolism pathways were activated earlier in PWN-resistant *P. thunbergii* (1 d vs. 3 d) than in susceptible *P. thunbergii* (3 d vs. 7 d), while the “biosynthesis of various secondary metabolites—part 2” pathway was activated later in PWN-resistant *P. thunbergii* (3 d vs. 7 d) than in susceptible *P. thunbergii* (1 d vs. 3 d). In addition, the MAPK signaling pathway was active for longer in resistant *P. thunbergii*. In general, the findings that PWN-resistant *P. thunbergii* activated additional pathways to suppress PWN infection and temporal changes of gene expression involved in the defense response to PWN inoculation in susceptible and resistant *P. thunbergii* were clearly demonstrated. Further analysis was based on these different pathways and unigene data sets.

### 2.3. The qPCR Validation

High-quality RNA (OD 260/280 = 1.8~2.2, RIN ≥ 7.9) was used to generate a sequencing library (Appendix A). To validate the results of the RNA-Seq data, seven DEGs (Appendix A) were selected to assess their expression by qPCR. The unigenes selected for qRT-PCR analysis were mainly involved in ROS-responsive genes, resistance proteins, signal transduction and membrane components. The qPCR assay for the selected transcripts shows an expression pattern similar to that obtained by RNA-Seq data analysis. The differential expression detected by RT-PCR was consistent with the RNA-Seq expression profiles of each of the selected transcripts (Figure 3).

### 2.4. The Alpha-Linolenic Acid and Linoleic Acid Metabolism Were Involved in the Response to PWN

In this study, alpha-linolenic acid and linoleic acid metabolism were significantly different in susceptible (*p* < 0.05) and resistant *P. thunbergii* (*p* < 0.001), and they were activated earlier in resistant *P. thunbergii* (Figure 4A). In the linoleic acid metabolism, the expression of *lindoleate 9S-lipoxygenase* (*LOX1-5*) and *13 s-lipoxygenase* (*LOX2S*) showed significant differences between susceptible and resistant *P. thunbergii.* Three unigenes encoding *LOX1-5* were up-regulated in susceptible *P. thunbergii*. At the same time, more unigenes encoding *LOX1-5* were activated in resistant *P. thunbergii*, and we found that six of the seven unigenes encoding *LOX1-5* were up-regulated and one (*TRINITY_DN3608_c0_g1*) was down-regulated. The unigenes encoding *LOX2S* were up-regulated, and only identified in resistant *P. thunbergii*. In contrast to the susceptible phenotype, the *LOX2S*-catalyzed branch was activated in the resistant phenotype, in addition to the *LOX1-5*-catalyzed branch being strengthened.

In alpha-linolenic acid metabolism, five enzymes showed significantly different expression, namely, hydroperoxide dehydratase (AOS), 12-oxophytodienoic acid reductase (OPR), LOX1-5, alcohol dehydrogenase class P (ADH1) and jasmonate O-methyltransferase (E.2.1.1.141). Among these, OPR, ADH1 and jasmonate O-methyltransferase were down-regulated in resistant *P. thunbergiiat* stage R1, whereas AOS and LOX2S were up-regulated (Figure 4B). In susceptible *P. thunbergii*, only AOS and OPR were altered at the S2 stage. This suggested that PWN activated a completely different defense system in resistant *P. thunbergii*, involving more unigenes and metabolic branches in linoleic acid and alpha-linolenic acid metabolism.

### 2.5. Regulation of the “MAPK Signaling Pathway—Plant” and “Biosynthesis of Various Secondary Metabolites—Part 2” Contributed to PWN Resistance

The results showed that a total of 4 and 17 DEGs were identified in susceptible and resistant *P. thunbergii*, respectively (Appendix A). In the MAPK signaling pathway, the *disease resistance protein RPS2* (*SUMM2*), *pathogenesis-related genes* (*PR1*) and *catalase* (*CAT1*) were significantly up-regulated in susceptible *P. thunbergii* in the first stage of PWN infection (S1), whereas they were not significantly expressed in the second stage (S2) of PWN infection (Figure 5A). On the other hand, more DEGs were activated in resistant *P. thunbergii*, including *basic endochitinase B* (*CHIB*), *nucleoside-diphosphate kinase* (*NDPK2*), *abscisic acid receptor PYL familiar* (*PYL*) and *serine/threonine-protein kinase OXI1* (*OXI1*), in addition to *SUMM2*, *PR1* and *CAT1*. The expression of *SUMM2* and *CHIB* was significantly up-regulated, while that of *NDPK2* and *PYL* was significantly down-regulated. The *PR1* gene was up-regulated at the S1 stage in susceptible *P. thunbergii*, but was significantly (*p* < 0.001) up-regulated at both the R1 and R2 stages in resistant *P. thunbergii* (Figure 5B).

In the biosynthesis pathway of various secondary metabolites, one unigene encoding *PLR* was down-regulated in susceptible *P. thunbergii*, while three unigenes encoding *PLR* were down-regulated in resistant *P. thunbergii* (Appendix A), and the unigene expression trends were completely opposite (Figure 5C,D). These data strongly suggested that substantial changes in defense pathways were activated by PWN between the susceptible and resistant *P. thunbergii*, and that more genes and pathways were activated to defend against PWN infection in resistant *P. thunbergii*. However, the specific resistance function of DEGs needs to be further investigated.

### 2.6. Characterization of Gene Expression Related to Terpenoids Biosynthesis

In this study, DEGs encoding terpenoids were the most abundant, whereas DEGs encoding sesquiterpenes and triterpenes were not detected. In addition, the number of terpene-encoding DEGs was lower and decreased continuously with the PWN infection (Figure 6A). This suggested that terpenoids might play an important role in the response of *P. thunbergii* to PWN infection. Another result supported this conclusion. We found that three terpenoid synthesis pathways were enriched in resistant *P. thunbergii* by PWN infection, namely, monoterpenoid biosynthesis (*p* = 0.0136), diterpenoid biosynthesis (*p* < 0.001) and terpenoid backbone biosynthesis (*p* = 0.0628) (Appendix A). And only the diterpenoid biosynthesis pathway was enriched in both susceptible and resistant *P. thunbergii*. Nine DEGs involved in the diterpenoid biosynthesis were found in susceptible and resistant *P. thunbergii*. Among these DEGs, the expression levels of *ent-copalyl diphosphate synthase* (E5.5.1.13), *(13E)-labda-7,13-dien-15-ol synthase* (E3.1.7.10) and *gibberellin 2beta-dioxygenase* (E1.14.11.13) were significantly different (Figure 6B). The *ent-copalyl diphosphate synthase* was down-regulated in the resistant phenotype at the R1 stage, but significantly up-regulated in the susceptible *P. thunbergii* at the S2 stage. Another synthase with *geranylgeranyl-pp* (*GGPP*) as substrate varied in susceptible and resistant *P. thunbergii*. In addition, *gibberellin 2beta-dioxygenase* was only activated in the resistant phenotype. This suggested that terpenoids were involved in the resistance of *P. thunbergii* to PWN.

## 3. Discussion

Breeding of pines for resistance to PWN is an effective means for controlling PWD. Currently, it is rarely possible to obtain somatic pine plants from selected elite material. Fortunately, our group had previously obtained some somatic plants of *P. thunbergii* from resistant material [28,40]. In this study, we wanted to investigate the differences at the transcriptome level of the black pine stem segment between resistant and susceptible trees inoculated with PWN. We found that there was a remarkable difference (*p* < 0.05) between the two phenotypes’ stem transcriptomes; the resistant phenotype activated more DEGs in response to PWN invasion, and the number of DEGs was always more than for the susceptible phenotype at each PWN infection stage (Figure 1). This was consistent with previous study of *P. thunbergii* inoculation with PWN [16]. However, in the study of *P. massoniana*, the number of DEGs in the resistant phenotype was always lower than in the susceptible phenotype at each time point [38]. This suggested that the response mechanisms to PWN infection were different between *P. massoniana* and *P. thunbergii*, possibly due to their different susceptibility, and therefore in-depth studies on susceptible pines were needed.

According to the GO and KEGG classification of DEGs, the differences in the significant (*p* < 0.05) GO terms and metabolic pathways between the susceptible and resistant *P. thunbergii* indicated qualitative and quantitative differences in the defense genes induced in response to PWN infection (Table 1 and Figure 2). A proportion of the DEGs in susceptible trees were involved in stress/defense response categories, such as the response to stimulus, biological regulation and transcriptional regulator activity. Furthermore, a large percentage of DEGs were categorized as involving the cellular process, metabolic process, membrane part, catalytic activity and binding. These GO terms were all enriched in the resistant phenotype and many more DEGs were involved in it. This suggested that the PWN-resistant *P. thunbergii* has developed sophisticated defense mechanisms to combat PWN invasion, such as blocking pathogen entry and activating a range of defense responses.

The alpha-linolenic acid metabolism and linoleic acid metabolism were, for the first time, intensively observed as to their functions against PWN infection in pines. In our present study, alpha-linolenic acid metabolism and linoleic acid metabolism were activated earlier in resistant *P. thunbergii* (R1 stage) than in susceptible *P. thunbergii* (S2 stage) after inoculation with PWN. Alpha-linolenic acid can inhibit innate immune responses associated with callose deposition in wheat [41], and the activated alpha-linolenic acid metabolic pathway has resulted in increased plant (*Citrus junos*) growth [42]. The genes *LOX*, *AOS* and *OPR* were also important for plant defenses against biotic and abiotic stresses in alpha-linolenic acid metabolism [43,44]. In our study, the JA-related synthetic enzymes *LOX2S* and *AOS* were up-regulated in both resistant and susceptible *P. thunbergii*, but the expression of *OPR* was exactly opposite between resistant (down-expression) and susceptible (up-expression) *P. thunbergii*. In the *P. pinaster* study, the expression levels of *AOS* and *LOX2S* were consistent with our results, but *OPR* was also up-regulated in susceptible trees, contrary to our results [37] The differential expression of *OPR* affected the downstream regulation of JA. Also, the JA played an essential role in the response to PWN infection and might be important for resistance [45]. Alpha-linolenic acid metabolism was known to be a precursor of JA biosynthesis [46,47]. Interestingly, the *jasmonate O-methyltransferase* (E2.2.1.141) was down-regulated in resistant *P. thunbergii* in our study, which would lead to a decrease in methyl-jasmonate. Rodrigues et al. [48] reported higher levels of methyl-jasmonate in susceptible trees in response to PWN in *P. pinaster* at 72 dpi. Hormone quantification showed that JA levels were significantly higher in inoculated *P. pinaster* compared to controls observed at the same time [37] Moreover, methyl-jasmonate promoted chlorophyll degradation in plants such as *Chenopodium album* [49] and apple fruit [50]. Therefore, we hypothesized that down-regulated expression of the *jasmonate O-methyltransferase* was one of the key factors for delayed wilting in resistant *P. thunbergii*. These results implied that the conversion of JA to methyl jasmonate had an important effect on the resistance of pines to PWN. However, their molecular mechanisms in response to the PWN infection need to be further investigated. Taken together, these results provide evidence that PWN inoculation can promote JA signaling by stimulating alpha-linolenic acid metabolism and linoleic acid metabolism to protect pines from pathogen infection.

MAPK signaling was one of the earliest responses following pathogen attack and played a key role in plant signaling in response to various stresses [51]. In this study, the “MAPK signaling pathway—plant” was significantly activated in both resistant (*p* < 0.001) and susceptible (*p* < 0.05) *P. thunbergii* (Table 1). Some genes involved in plant hypersensitive response (HR) and disease resistance proteins were enriched. The resistance protein *SUMM2* was significantly up-regulated in expression during the first stage (R1 and S1) in both susceptible and resistant *P. thunbergii* infection with PWN. In addition, the *PR1* was only significantly (*p* < 0.05) up-regulated at the S1 stage in susceptible phenotypes, but in resistant phenotypes, the *PR1* was significantly (*p* < 0.001) up-regulated at both the R1 and R2 stages. In plant–pathogen interactions, resistance proteins recognized pathogen effectors and then initiated effector-triggered immunity, which was often associated with (HR) cell death [52,53]. The resistance protein SUMM2 mediated immunity and HR cell death [51,54]. HR cell death at the site of infection was critical for initiating systemic signals that activate distal plant immunity and ultimately lead to systemic acquired resistance (SAR) [55,56]. In addition, HR cell death has often been described as an immune strategy for the prevention of pathogen colonization through the recognition of adapted biotrophic or hemi-biotrophic pathogens [57,58]. Therefore, we proposed that *SUMM2* could be an important resistance gene functioning to prevent further spread of the pathogen in *P. thunbergii* during the early stages of PWN infection. Furthermore, PR1 was an important defense protein which was commonly used as a marker for SAR [59]. Plant PR proteins were initially identified as proteins that were strongly induced under biotic and abiotic stresses [60,61]. The mode of action of most PR proteins has been well characterized, with the exception of PR1, which belongs to a large superfamily of proteins which shares a common CAP structural domain. The importance of these proteins in immune defense was illustrated by the fact that PR1 overexpression in plants resulted in increased resistance to pathogens [62]. The defense response of the PR gene (*PR1-10*) to PWN has been reported in previous studies in *P. thunbergii*, and *PR1* was predominantly up-regulated for expression at the early infestation stage (1 and 3 dpi) (Hirao et al., 2012). This is consistent with our findings. Our results indicated that *SUMM2* and *PR-1* were induced by PWN in both susceptible and resistant trees for plant immunity associated with HR cell death, and the *PR1* worked longer in the resistant phenotype defending against the PWN infection. Furthermore, we found that comparatively more DEGs involved in ROS mediation were activated by the resistant phenotype in response to PWN infection, such as *CAT1*, *NDPK2*, *OXI1*, whereas only *CAT1* was identified as a gene regulating ROS in the susceptible phenotype. Harao, et al. [16] reported that the *peroxidase* (*PR-10*) were the characteristic significantly up-regulated genes in resistant *P. thunbergii* inoculation with PWN. In the *P. massoniana* study, the *catalase*, *peroxidase*, *superoxidase dismutase* and *glutathione reductase* genes were involved in the defense response to PWN. In comparison to previous reports in *P. thunbergii*, we have identified more antioxidant-related genes of resistant *P. thunbergii* functioning against PWN. And compared to the more non-susceptible *P. massoniana*, both resistant trees had more complex ROS regulation relative to susceptible trees, although the antioxidant genes were not exactly the same. These results implied that ROS scavenging capacity was closely related to pine defense against PWN infestation. ROS-related genes regulating the resistance of pines to PWN infestation will be a topic worthy of further study.

In the “biosynthesis of various secondary metabolites—part 2” pathway, the *pinoresinol/lariciresinol reductase* (*PLR*) marked a significant difference between resistant and susceptible *P. thunbergii*. The resistant phenotype activated more DEGs involved in *PLR* regulation and had a different gene expression pattern than the susceptible phenotype (Figure 5). The *PLR* catalyzed the sequential reduction of pinoresinol to secoisolariciresinol via lariciresinol, which could lead to the structural and stereochemical diversity of lignans [63]. Lignin also affected the mechanical strength of the cell wall, which has been considered the first physical line of defense against PWN infection [9,64]. A more differential expression of the *PLR* might be associated with the regulation of lignin in response to PWN infection in resistant *P. thunbergii*, suggesting that the resistant phenotype had a more complicated regulation of lignin as a physical defense to suppress PWN infection.

Terpenoid metabolism has played an important role in the resistance of pines to PWN [38]. In our study, the DEGs encoding diterpenoids were the most numerous, and the diterpenoid biosynthetic pathway was significantly enriched in resistant *P. thunbergii*. After the trunk of a conifer suffered an insect attack, pathogen invasion or mechanical wounding, oleoresin could be synthesized [65,66]. Monoterpenes, sesquiterpenes and diterpene resin acids are important components of oleoresin [67,68]. Monoterpenes and sesquiterpenes in turpentine oil could directly affect herbivores through the release of toxic volatiles, and diterpene resin acids form physical barriers at the wound site [68]. In *P. thunbergii*, diterpenoids have the most DEGs. Therefore, we suggest that the diterpenoids metabolism is the main mode of response to PWN infestation. Furthermore, the *ent-copalyl diphosphate synthase* and *gibberellin 2beta-dioxygenase* unigenes were significantly differentially expressed in the diterpenoid biosynthetic pathway in this study. *Gibberellin 2beta-dioxygenase*, a key gene in the gibberellin synthesis, was only activated in resistant *P. thunbergii*. Tanaka, et al. [69] reported that *gibberellin 2beta-dioxygenase* increased rice biomass under low-nutrient conditions. Prisic, et al. [70] reported that *ent-copalyl diphosphate synthases* (*OsCPS1ent* and *OsCPS2ent*) from rice (*Oryza sativa*) were involved in GA biosynthesis and related a secondary metabolism producing defensive phytochemicals. In our study, the resistant *P. thunbergii* positively regulated GA signaling after inoculation with PWN, suggesting that GA biosynthesis genes were involved in the resistance of *P. thunbergii* to PWN.

## 4. Materials and Methods

### 4.1. Plant Material

In this experiment, the three-year-old susceptible *P. thunbergii* and somatic plants of PWN-resistant *P. thunbergii* were used as experimental materials. The somatic plants were obtained from the 1539-1 cell line by somatic embryogenesis (Sun et al., 2019). The 1539-1 cell line was initiated from the resistant family 39 of *P. thunbergii*. We collected immature cones (open pollination, family 39) from the seed orchard, and used their female gametophyte to induce embryogenic tissue (1539-1) (Sun et al., 2019). The somatic embryos were obtained and germinated into somatic plants, which were then transplanted in field (Sun et al., 2023); these somatic plants were the experimental material for this paper. The susceptible *P. thunbergii* (mortality rates were all 100% after inoculation with PWN at 28 days in the previous tests) from Suqian City, China, was selected as the control.

### 4.2. Pine Wood Nematode Inoculation and Sampling

The three-year-old susceptible and resistant *P. thunbergii* trees were transplanted into pots and grown under the same conditions for three months, and then the plants were transferred to a forcing house (30 °C constant temperature) for inoculation with PWN. The PWN used in our study was the highly virulent isolate AMA3. Three branches per tree were inoculated (1000 nematodes per branch) in susceptible and resistant *P. thunbergii* trees. Samples were collected at 1, 3, 7 and 14 days post-inoculation (dpi) to assess the somatic plants’ resistance to PWD. We sampled inoculated trees (both susceptible and resistant trees) at 1, 3 and 7 dpi; resistant trees were also sampled at 14 dpi. The last sampling time was chosen based on previous results, in which the needles of resistant and susceptible *P. thunbergii* turned yellow at 14 and 7 dpi, respectively, which was also observed in this experiment. The 2 cm long segments of the stems below the inoculation sites were cut off and immediately placed in liquid nitrogen. These samples were then stored at −80 °C for further RNA extraction. Three trees representing resistant and susceptible *P. thunbergii* were selected as biological replicates for each treatment.

### 4.3. RNA Isolation Procedure and Quantification

Following inoculation of resistant and susceptible *P. thunbergii* with PWN, samples were collected at the critical time point of disease development and stored at −80 °C as a reserve. Total RNA was extracted from each PWN-inoculated tree sample at four time points using TRIzol^®^ Reagent (plant RNA purification reagent for plant tissue) according to the instructions, while genomic DNA was removed using DNase I (TaKara, Nanjing, China). The RNA degradation and contamination was monitored using 1% agarose gels, while the integrity and purity of the total RNA quality was determined using a 2100 Bioanalyser (Agilent Technologies, Shanghai, China) and quantified using an ND-2000 (NanoDrop Technologies, Wilmington, DE, USA). Only high-quality RNA samples (OD 260/280 = 1.8~2.2, RIN ≥ 7.9) were used to develop a sequencing library.

### 4.4. Quantitative Real-Time PCR Analysis

The RNA samples used for the qRT-PCR and transcriptome sequencing were identical. To evaluate the accuracy and reproducibility of the RNA-Seq expression profiles, quantitative real-time polymerase chain reaction (qRT-PCR) analysis was performed to analyze the expression levels of regulated genes at four different time points. Quantitative RT-PCR was run on a 7900 Real Time PCR System (Applied Biosystems, California USA) using the SYBR Green detection method to verify the transcriptome sequencing results. Quantitative real-time PCR (qRT -PCR) was performed in a 20 μL reaction volume containing 10 μL of SYBR Green Master Mix (Vazyme Biotech, Nanjing, China).

### 4.5. De Novo Assembly and Annotation

The raw paired end reads were trimmed and quality control was performed using fastp with default parameters. The clean data from the samples were then used to perform de novo assembly using Trinity. The assembled transcripts were then evaluated and optimized using BUSCO protein groups and the GO and KEGG databases. BLASTX was used to identify the proteins with highest sequence similarity to the given transcripts to retrieve their functional annotations, while typical cut-off E-values of less than 1.0 × 10^−5^ were set.

### 4.6. Differential Expression Analysis and Functional Enrichment

To identify the DEGs (differential expression genes) between the resistant and susceptible *P. thunbergii* samples, the expression levels of each gene were calculated using the transcripts-per-million-reads (TPM) method. RSEM (http://deweylab.biostat.wisc.edu/rsem/, accessed on 20 September 2022) was used to quantify gene abundance. Differential expression analysis was performed using the DESeq2/DEGseq/edgeR/Limma/, where DEGs with |log2 (foldchange)| ≥ 1 and *P*-adjust ≤ 0.05 were considered to be significant. Furthermore, functional-enrichment analysis, including GO (*Gene Ontology*, http://www.geneontology.org, accessed on 8 January 2023) and KEGG (*Kyoto Encyclopedia of Genes and Genomes*, http://www.genome.jp/kegg/, accessed on 8 January 2023) was also performed to identify which of the DEGs were significantly enriched in GO terms. Metabolic pathways were then compared to the whole-transcriptome background at *p* ≤ 0.05.

### 4.7. Statistical Analysis

Data were analyzed using analysis of variance (ANOVA) SPSS19 software (SPSS Inc., Chicago, IL, USA). Results were expressed as percentages using nonparametric methods. In addition, the graphs were created using Prism (Patterns & Practices., Redmond, WA, USA) and Adobe Photoshop CS6 (64 bit) software (Adobe, San Jose, CA, USA).

## 5. Conclusions

This work was the first to use resistant *P. thunbergii* somatic plants to study the defense responses of *P. thunbergii* to PWN; it promotes the use of somatic plants in the future. In this study, the resistant *P. thunbergii* activated more genes to defend against the PWN inoculation. Resistance protein *SUMM2*, *PR1* and ROS-related genes were significantly up-expressed to contribute to the protection of *P. thunbergii* against PWN inoculation. In addition, alpha-linolenic acid metabolism and linoleic acid metabolism contributed to PWN resistance by regulating the JA-related synthetic enzyme genes. In conclusion, this study provided the first comprehensive report of the somatic plant transcriptomes of *P. thunbergii* materials which are resistant in response to PWN, and it enhanced the genetic resource database for *P. thunbergii*.

## Figures and Tables

**Figure 1 ijms-24-14376-f001:**
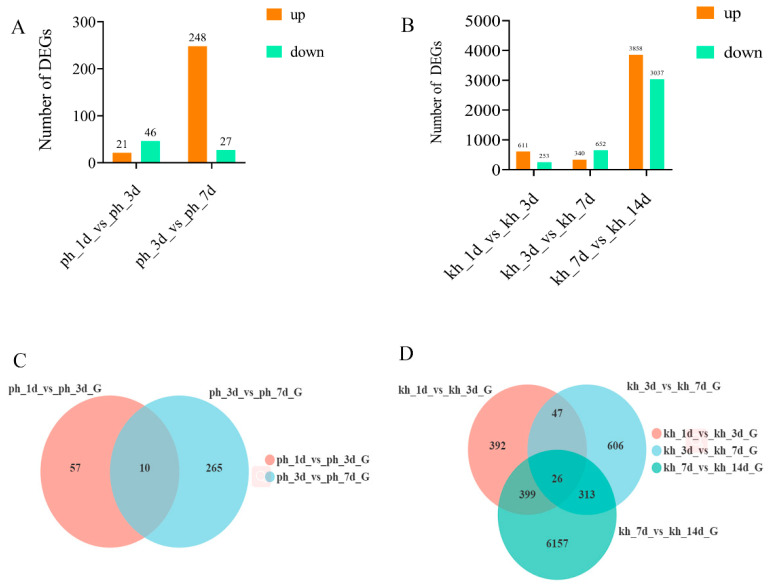
Comparison of DEGs between susceptible and resistant *P. thunbergii* at different disease stages. (**A**,**B**) indicated the number of DEGs obtained in susceptible and resistant *P. thunbergii* at infection stages, respectively. (**C**,**D**) indicated the number and overlapping relationships of DEGs in venn diagram between susceptible and resistant *P. thunbergii* at different infection stages, respectively.

**Figure 2 ijms-24-14376-f002:**
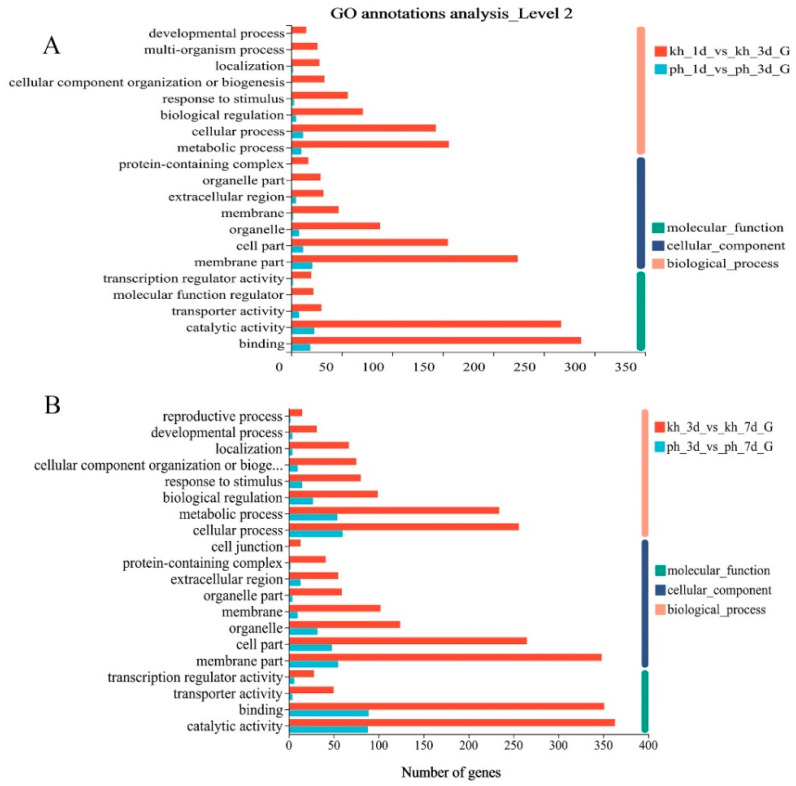
Enriched biological processes in DEGs by GO annotation for susceptible and resistant *P. thunbergii* at different infection stages. (**A**) indicated GO annotation of susceptible and resistant *P. thunbergii* at first infection stage (1d vs 3d). (**B**) indicated GO annotation of susceptible and resistant *P. thunbergii* at second infection stage (3d vs 7d).

**Figure 3 ijms-24-14376-f003:**
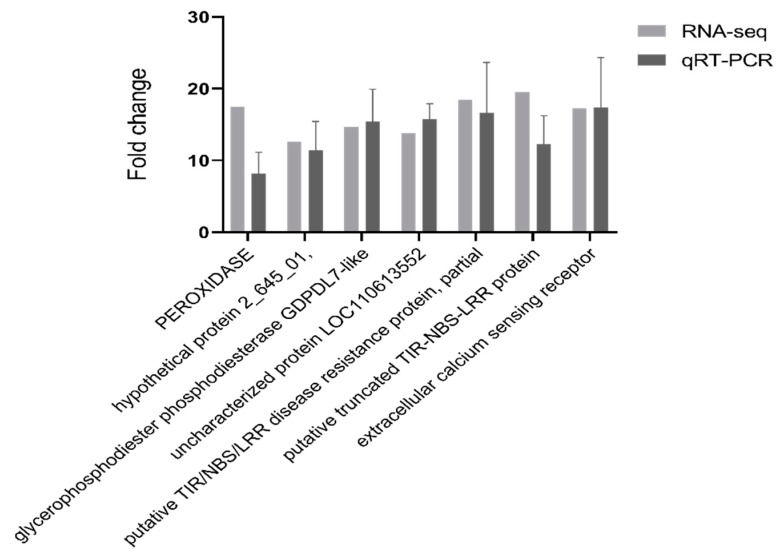
qRT-PCR validation of selected transcripts for validation. Relative expression levels of qRT-PCR are calculated using elongation factor 1-alpha as the internal control. The data are expressed as the mean (±SE). Error bars represent the SE.

**Figure 4 ijms-24-14376-f004:**
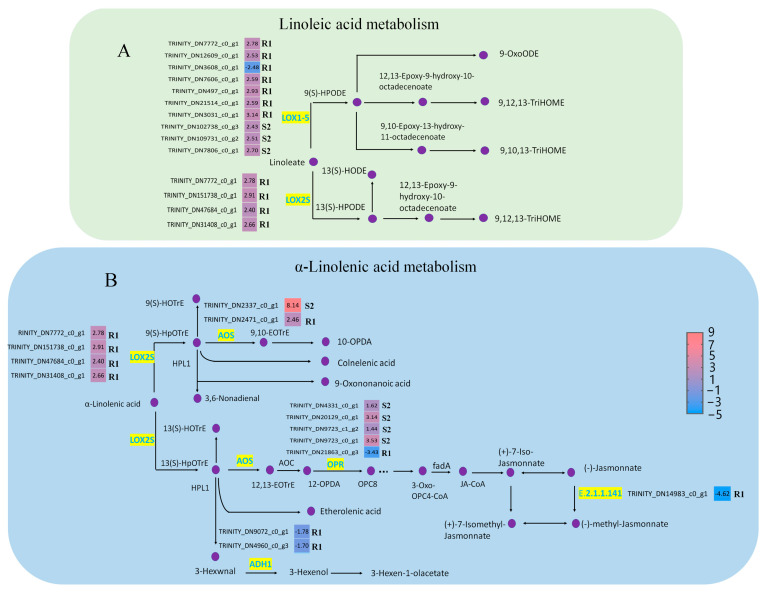
DEGs involved in alpha-linolenic acid metabolism and linoleic acid metabolism in *P. thunbergii*. (**A**) indicated linoleic acid metabolism pathway. (**B**) indicated alpha-linolenic acid metabolism pathway. Enzymes involved in each step are shown in purple, and the green boxes represent DEGs encoding enzyme activity. R represents resistant *P. thunbergii*. S represents susceptible *P. thunbergii*. R1 represents the first stage of the resistant *P. thunbergii* inoculated with PWN (1 d vs. 3 d). S2 represents the second stage of susceptible *P. thunbergii* inoculated with PWN (3 d vs. 7 d). *LOX1-5* (E5.5.1.13) represents *lindoleate 9S-lipoxygenase*. *LOX2S* (E1.14.11.13) represents *lipoxygenase*. *AOS* represents *hydroperoxide dehydratase*. *OPR* represents *12-oxophytodienoic acid reductase*. *ADH1* represents *alcohol dehydrogenase class-P*, E2.2.1.141 represents *jasmonate O-methyltransferase*.

**Figure 5 ijms-24-14376-f005:**
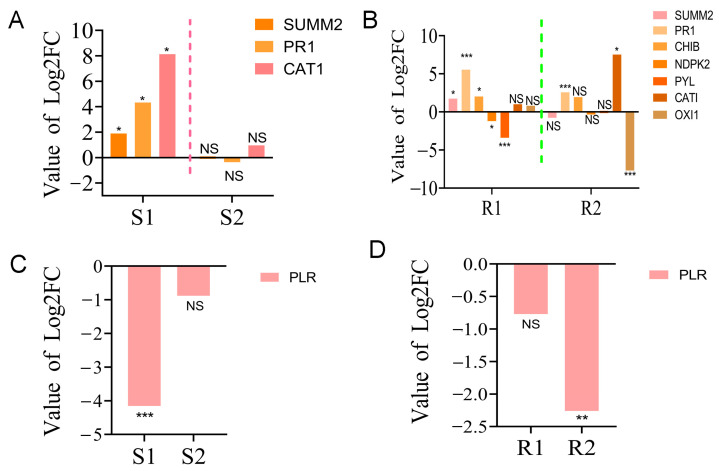
The expression of partial DEGs in “MAPK signaling pathway—plant” and “biosynthesis of various secondary metabolites—part 2” pathways. (**A**,**B**) The expression of DEGs in the “MAPK signaling pathway—plant” in susceptible and PWN-resistant *P. thunbergii*, respectively. (**C**,**D**) Expression of DEGs in “biosynthesis of various secondary metabolites—part 2” in susceptible and PWN-resistant *P. thunbergii*, respectively. S1 represents the 1st stage (1 d vs. 3 d) of PWN infection in susceptible *P. thunbergii*. S2 represents the 2nd stage (3 d vs. 7 d) of PWN infection in susceptible *P. thunbergii*. R1 represents the 1st stage (1d vs. 3d) of PWN infection in resistant *P. thunbergii*. R2 represents the 2nd stage (3 d vs. 7 d) of PWN infection in resistant *P. thunbergii*. * and ** indicate significant differences at *p* < 0.05 and *p* < 0.01, respectively. *** indicate significant differences at *p* < 0.001.

**Figure 6 ijms-24-14376-f006:**
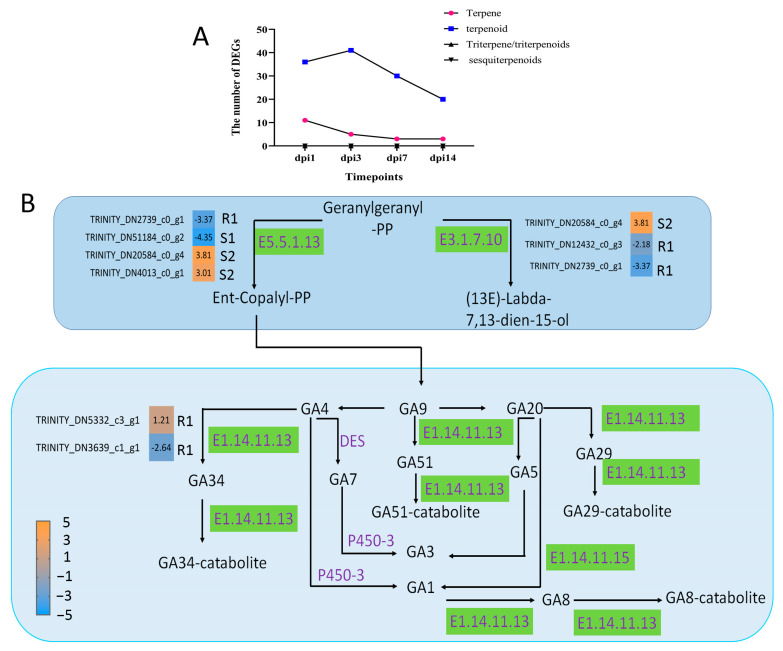
DEGs involved in diterpenoid biosynthesis in the stem of *P. thunbergii*. (**A**) indicates the number of DEG terpenoid types. (**B**) indicated diterpenoid biosynthesis pathway. Enzymes involved in each step are shown in purple, and the green boxes represent DEGs encoding enzyme activity. R represents resistant *P. thunbergii*. S represents susceptible *P. thunbergii*. R1 represents the first stage of the resistant *P. thunbergii* inoculated with PWN (1 d vs. 3 d). S1 represents the first stage of the susceptible *P. thunbergii* inoculation with PWN (1 d vs. 3 d). S2 represents the second stage of susceptible *P. thunbergii* inoculated with PWN (3 d vs. 7 d). E5.5.1.13 represents *ent-copalyl diphosphate synthase*. E3.1.7.10 represents *(13E)-labda-7,13-dien-15-ol synthase*. E1.14.11.13 represents *gibberellin 2beta-dioxygenase*.

**Table 1 ijms-24-14376-t001:** The KEGG analysis of DEGs in susceptible and resistant *P. thunbergii*.

Host of PWN	1 d vs. 3 d		3 d vs. 7d	
	Pathway Description	*p*-Value	Pathway Description	*p*-Value
Susceptible *P. thunbergii*	MAPK signaling pathway—plant	0.0053	alpha-Linolenic acid metabolism	0.0064
	Biosynthesis of various secondary metabolites—part 2	0.0239	Linoleic acid metabolism	0.0153
Resistant *P. thunbergii*	MAPK signaling pathway—plant	*p* < 0.001	Cutin, suberine and wax biosynthesis	*p* < 0.001
	Linoleic acid metabolism	*p* < 0.001	MAPK signaling pathway—plant	*p* < 0.001
	Plant–pathogen interaction	*p* < 0.001	Plant–pathogen interaction	*p* < 0.001
	Plant hormone signal transduction	*p* < 0.001	Photosynthesis—antenna proteins	*p* < 0.001
	alpha-Linolenic acid metabolism	*p* < 0.001	Phenylpropanoid biosynthesis	*p* < 0.001
	Flavonoid biosynthesis	*p* < 0.001	Biosynthesis of various secondary metabolites—part 2	*p* < 0.001
	Diterpenoid biosynthesis	0.005	Photosynthesis	0.015
	Protein processing in endoplasmic reticulum	0.018		

## Data Availability

The data described in this study can be found in the article and the Appendix A.

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
