# Peer review of "Resistant and Susceptible Pinus thunbergii ParL. Show Highly Divergent Patterns of Differentially Expressed Genes during the Process of Infection by Bursaphelenchus xylophilus"

_ijms, 2023, doi:10.3390/ijms241814376_

Round 1

Reviewer 1 Report

Here are my comments:

- In abstract and conclusion, highlight 1-2 key findings rather than summarizing all results.

- The introduction would benefit from more background on the global impact and significance of pine wilt disease, and the importance of identifying resistance mechanisms in pine trees. 

- Expand on why transcriptomic analysis of resistant somatic pine plants is a useful approach to shed light on resistance mechanisms.

- In results section, highlight the most novel findings first before describing secondary results.

- Provide more biological interpretation of the gene expression changes - what do they mean functionally? How do they confer resistance?

- Elaborate on how metabolic changes like terpenoid and fatty acid metabolism influence resistance. Explain mechanisms.

- Discuss how findings compare to transcriptomic studies in other pine-nematode pathosystems. What new insights were gained here? 

- Expand discussion of implications - how could results inform future breeding or diagnostic efforts?

- Carefully edit for clarity, prose, grammar, formatting, style consistency. Define all genes/abbreviations.

- Supplement with additional data files/tables as evidence to support claims.

- Reorganize order of figures/tables for logical flow. 

Author Response

Dear editors and reviewers

Thank you very much for your comments and professional advice. Base on your suggestions and requests, we have made corrected modifications on the revised manuscript. Modified sections are highlighted in yellow. The details as follows:

reviewer comments:

Reviewer 1#

  1. In abstract and conclusion, highlight 1-2 key findings rather than summarizing all results.

Response: The abstract (Page 1 and 2) and conclusion (Page 13 and 14) section have been rewritten as you suggested.

  1. The introduction would benefit from more background on the global impact and significance of pine wilt disease, and the importance of identifying resistance mechanisms in pine trees.

Response: In introduction section, more background on the global impact and significance of pine wilt disease was added. Page 2, line 43-55. And the importance of identifying resistance mechanisms in pine trees. Page 2, line 69-77.

  1. Expand on why transcriptomic analysis of resistant somatic pine plants is a useful approach to shed light on resistance mechanisms.

Response: The importance of transcriptomic analysis on somatic pine plants resistance to nematode was added in text. Page 2, line 78-81.

  1. In results section, highlight the most novel findings first before describing secondary results.

Response: In the results section, we reordered our findings. Page 6-10.

  1. Provide more biological interpretation of the gene expression changes - what do they mean functionally? How do they confer resistance?

Response: We expand on the biological interpretation of the gene expression changes in the discussion section. Page 10, line 288-300 and page 11, line 328-347.

  1. Elaborate on how metabolic changes like terpenoid and fatty acid metabolism influence resistance. Explain mechanisms.

Response: We have added an exploration of fatty acid metabolism (Page 100, line 285-300) and terpenoid (Page 12, line 375-385) influence resistance in the Discussion section.

  1. Discuss how findings compare to transcriptomic studies in other pine-nematode pathosystems. What new insights were gained here?

Response: The new insights were gained in the discussion section was added in text. Page 10, line 285-286, line 304-316.

  1. Expand discussion of implications - how could results inform future breeding or diagnostic efforts?

Response: The impact of the results on future breeding and disease diagnosis is discussed. Page 1, line 26-28.

  1. Carefully edit for clarity, prose, grammar, formatting, style consistency. Define all genes/abbreviations.

Response: The manuscript has been further revised as per your suggestions. In addition, the full names of all genes/abbreviations are defined in the text.

  1. Supplement with additional data files/tables as evidence to support claims.

Response: The additional data files were added in supplementary file.

  1. Reorganize order of figures/tables for logical flow.

Response: In the results section, we reordered our findings. Page 6-10.

Reviewer 2 Report

The study was focused on resistant and susceptible Pinus thunbergii that exhibited highly divergent patterns of differentially expressed genes during the infection by Bursaphelenchus xylophilus. The study is quite interesting, however, I recommend some important revisions:

-        Materials an methods: RNA isolation procedure and quantification of its purity and concentration, as well as detailed description of transcriptomic analyses and validation of transcriptomic analyses by real-time qRT-PCR should be added. Statistical analyses paragraph should be included.

-        Results: I recommend including the electropherograms presenting the RNA bands in agarose gels in the manuscript or in the Supplementary file – it would provide information regarding quality of total RNA samples. The RNA concentration should be quantified at both 260 and 280 nm, and purity of RNA should be calculated based on the A260/A280 coefficient.

-        Results: I recommend validation of transcriptomic analyses by real-time qRT-PCR (approx. 5-10 randomly selected genes). In such case, the house-keeping gene, Genbank accession numbers of tested genes, chemicals, real-time qRT-PCR assay, should be described in Materials and methods section.

-        Moderate editing of English language is required.

 Moderate editing of English language is required.

Author Response

Dear editors and reviewers

Thank you very much for your comments and professional advice. Base on your suggestions and requests, we have made corrected modifications on the revised manuscript. Modified sections are highlighted in yellow. The details as follows:

reviewer comments:

Reviewer2#

The study was focused on resistant and susceptible Pinus thunbergii that exhibited highly divergent patterns of differentially expressed genes during the infection by Bursaphelenchus xylophilus. The study is quite interesting, however, I recommend some important revisions:

  1. Materials and methods: RNA isolation procedure and quantification of its purity and concentration, as well as detailed description of transcriptomic analyses and validation of transcriptomic analyses by real-time qRT-PCR should be added. Statistical analyses paragraph should be included.

Response: In material and method section, the detailed descriptions of the RNA isolation process and its purity and concentration quantification, as well as transcriptome analysis and real-time qRT-PCR to validate transcriptome analysis have been added. Page 13, line 423-441. In addition, the statistical analysis paragraph was added. Page 13, line 460-464.

  1. Results: I recommend including the electropherograms presenting the RNA bands in agarose gels in the manuscript or in the Supplementary file – it would provide information regarding quality of total RNA samples. The RNA concentration should be quantified at both 260 and 280 nm, and purity of RNA should be calculated based on the A260/A280 coefficient.

Response: The RAN electrophoretic detection plots and quality assessment tables was added as the supplementary materials.

  1. Results: I recommend validation of transcriptomic analyses by real-time qRT-PCR (approx. 5-10 randomly selected genes). In such case, the house-keeping gene, Genbank accession numbers of tested genes, chemicals, real-time qRT-PCR assay, should be described in Materials and methods section.

Response: The information on transcriptomic analyses by real-time qRT-PCR was added in result section. Page 6, line 156-168.

  1. Moderate editing of English language is required.

Response: The manuscript with linguistic revision by co-author Mati and Dr Frank.

Round 2

Reviewer 2 Report

The revision was properly concucted. The manuscript may be considered for publication. 

 Minor editing of English language required